Estimating average wind speed in Thailand using confidence intervals for common mean of several Weibull distributions

La-ongkaew Manussaya
Niwitpong Sa-Aat sa-aat.n@sci.kmutnb.ac.th
Niwitpong Suparat
Department of Applied Statistics, Faculty of Applied Science, King Mongkut’s University of Technology North Bangkok , Bangkok , Thailand
Kuriqi Alban
Electronic publication date: 2023 Jun 22
Publication date: 2023
Volume: 11
Electronic Location ID: e15513
Received 2022 Sep 11; Accepted 2023 May 15
Copyright: ©2023 La-ongkaew et al.
Copyright year: 2023
Copyright holder: La-ongkaew et al.
License: This is an open access article distributed under the terms of the Creative Commons Attribution License, which permits unrestricted use, distribution, reproduction and adaptation in any medium and for any purpose provided that it is properly attributed. For attribution, the original author(s), title, publication source (PeerJ) and either DOI or URL of the article must be cited.
License URL: https://creativecommons.org/licenses/by/4.0/

Keywords: Common mean, Bayesian method, Prior gamma distribution, Equitailed confidence interval, The highest posterior density, Generalized confidence interval, Adjusted method of variance estimates recovery, Simulation

Funding: King Mongkut’s University of Technology North Bangkok KMUTNB-66-KNOW-01 This research was funded by King Mongkut’s University of Technology North Bangkok. Grant No. KMUTNB-66-KNOW-01. The funders had no role in study design, data collection and analysis, decision to publish, or preparation of the manuscript.

==============================
The Weibull distribution has been used to analyze data from many fields, including engineering, survival and lifetime analysis, and weather forecasting, particularly wind speed data. It is useful to measure the central tendency of wind speed data in specific locations using statistical parameters for instance the mean to accurately forecast the severity of future catastrophic events. In particular, the common mean of several independent wind speed samples collected from different locations is a useful statistic. To explore wind speed data from several areas in Surat Thani province, a large province in southern Thailand, we constructed estimates of the confidence interval for the common mean of several Weibull distributions using the Bayesian equitailed confidence interval and the highest posterior density interval using the gamma prior. Their performances are compared with those of the generalized confidence interval and the adjusted method of variance estimates recovery based on their coverage probabilities and expected lengths. The results demonstrate that when the common mean is small and the sample size is large, the Bayesian highest posterior density interval performed the best since its coverage probabilities were higher than the nominal confidence level and it provided the shortest expected lengths. Moreover, the generalized confidence interval performed well in some scenarios whereas adjusted method of variance estimates recovery did not. The approaches were used to estimate the common mean of real wind speed datasets from several areas in Surat Thani province, Thailand, fitted to Weibull distributions. These results support the simulation results in that the Bayesian methods performed the best. Hence, the Bayesian highest posterior density interval is the most appropriate method for establishing the confidence interval for the common mean of several Weibull distributions.

Introduction

Greenhouse gases are produced by both natural processes and human activity, especially the burning of fossil fuels for electricity generation. Greenhouse gases have the ability to absorb infrared radiation, or heat radiation, that radiates off the surface of the earth. When there are large amounts of greenhouse gases, infrared radiation cannot be reflected back outside the atmosphere, causing the increased average global temperature and initiating extreme weather events. In 2017, Thailand ranked 20th in the world’s greenhouse gas emissions. It is located in the equatorial region, which is influenced by ocean currents that produce heavy rain and high wind speed during the monsoon season from mid-May to mid-October. These phenomena can be hazardous to both humans and animals, causing catastrophes that affect agricultural productivity, which is an important part of Thailand’s economy. The southern region of Thailand is a coastal area that is influenced by the southeast monsoon winds, and Surat Thani is a province on the southeastern coast of Thailand located on a peninsula that juts out into the sea. Thus, monitoring the wind speed to quantify and predict its potential intensity is a useful endeavor. Several distributions are suitable for studying wind speed data, which may differ depending on the month, season, and year. One of these is the Weibull distribution, which has been applied in several studies on analyzing wind speed. Genc et al. (2005) studied wind power potential by estimating two parameters of a Weibull distribution. Dokur & Kurban (2015) used the Weibull distribution to determine the wind energy potential in the Bilecik region and provided estimates of its parameters. Sasujit & Dussadee (2016) used the Weibull distribution to provide an assessment of wind energy and electricity generation in northern Thailand. Islam, Dussadee & Chaichana (2016) used it to estimate the wind power potential on Saint Martin’s Island in Bangladesh. La-ongkaew, Niwitpong & Niwitpong (2021) applied the coefficient of variation of the Weibull distribution to estimate the dispersion of wind speed data in Thailand. Shu & Jesson (2021) assessed the characteristics of wind speed datasets by using Weibull distributions. As well as assessing wind speed data, the Weibull distribution has been applied in studies in other areas. For illustration, it was utilized to assess the survival time of guinea pigs injected with varying doses of tubercle bacilli. (Bjerkedal et al., 1960), the failure times of air-conditioning systems in two airplanes (Proschan, 1963), the amounts of insurance claims (Hamza & Sabri , 2022), the shelf life of Pezik pickles (Keklik, Isikli & Sur , 2017), and the moisture content of milled rice (Ling, Teng & Lin, 2018).

The mean is a very important statistic for measuring the central tendency of a dataset and has been used in many applications; e.g., the amount of nitrogen-bound bovine serum albumin in mice (Hand et al., 1993; Schaarschmidt, 2013; Sadooghi-Alvandi & Malekzadeh, 2014), the amount of selenium in non-fat milk powder (Philip, Sun & Sinha, 1999), the CD4+ cell counts of HIV patients after initiating anti-vital therapy (Liang, Su & Zou, 2008), and rainfall in Chiang Mai, Thailand (Maneerat, Niwitpong & Niwitpong, 2019). Interval estimation for the mean of a distribution has been investigated by several research groups. Chen & Mi (1996) applied several maximum likelihood estimators for constructing the confidence interval for the mean of an exponential distribution based on grouped data. Colosimo & Ho (1999) estimated the confidence interval for the mean of a Weibull distribution for lifetime analysis based on censored reliability datasets. Peng (2004) provided estimates for the confidence interval for the mean of heavy-tailed distributions. Krishnamoorthy, Lin & Xia (2009) established estimates for the confidence interval for the mean of a Weibull distribution using the generalized variable approach and Wald confidence intervals. Thangjai, Niwitpong & Niwitpong (2020) applied Bayesian methodology to construct estimates of the confidence interval for the mean of a normal distribution with an unknown coefficient of variation. Maneerat, Nakjai & Niwitpong (2022) proposed using Bayesian noninformative priors to estimate the confidence interval for the mean of a three-parameter delta-lognormal distribution. Moreover, functions of the mean such as the difference between and the ratio of two means have also been reported. Lee & Lin (2004) used the generalized confidence interval (GCI) approach to estimate the confidence interval for the ratio of the means of two normal populations. Niwitpong & Niwitpong (2010) proposed estimates for the confidence interval for the difference between the means of two normal populations where the ratio of their variances is known. Niwitpong, Koonprasert & Niwitpong (2012) proposed estimates for the confidence interval for the difference between the means of several normal populations with known coefficients of variation. Thangjai, Niwitpong & Niwitpong (2017) used the GCI and large sample approaches to estimate the confidence interval for the mean and the difference between the means of several normal distributions with unknown coefficients of variation. Maneerat & Niwitpong (2020) compared medical care costs by using Bayesian intervals for the ratio of the means of several delta-lognormal distributions.

Since it is common practice to collect data in different settings, inference based on the common mean of several populations is a useful statistic. Indeed, many researchers have estimated the confidence interval for this scenario. Krishnamoorthy & Lu (2003) used the concept of the generalized p-value to estimate the confidence interval for the common mean of several normal populations. Lin & Lee (2005) proposed a generalized pivotal quantity (GPQ) using the best linear unbiased estimator for estimating the confidence interval of the common mean of several normal populations when the variances are unknown. Later, Ye, Ma & Wang (2010) provided interval estimation for the common mean when the scalar parameters among several inverse Gaussian populations have become unknown. Behboodian & Jafari (2014) used the GCI approach to determine the confidence interval for the common mean of several lognormal populations, while Smithpreecha, Niwitpong & Niwitpong (2018) proposed new methods to calculate the confidence interval for the common mean of several lognormal distributions based on the GCI and adjusted method of variance estimates recovery (MOVER) methods. Lin & Wu (2011) proposed an estimation method based on a higher-order likelihood-based procedure for the confidence interval for the common mean of several inverse Gaussian distributions. Maneerat & Niwitpong (2021) estimated the confidence interval for the common mean of several delta-lognormal populations using the fiducial GCI (FGCI), large sample, MOVER, parametric bootstrap, and highest posterior density (HPD) intervals using the Jeffreys’ rule or normal-gamma-beta prior.

In the present study, our goal was to compare the wind speed data from several locations to predict the occurrence of severe wind speed events. Since Surat Thani is a large province on the southeast coast of Thailand, using the common mean of the wind speed datasets from different areas will help in this endeavor, and thereby estimating the confidence interval for the common mean of several Weibull populations becomes important. The advantage of this study is that it will assist provincial authorities in estimating the amount of wind and predicting wind speed in order to monitor the occurrence of severe wind speed. Notwithstanding, the common mean of various Weibull populations has never been investigated. We used Bayesian methodology for the equitailed confidence interval and the HPD interval based on the gamma prior to estimate the confidence interval for the common mean of several Weibull distributions and compare their performances with GCI and adjusted MOVER. Furthermore, we applied these novel methods to assess real wind speed datasets from several locations in Surat Thani province, Thailand. Furthermore, there is no previous study on the implementation of their methodology for measuring the common mean of wind speed data. To fill the gap, novel methods for the confidence interval for the common mean of Weibull distributions were proposed by contemplating the wind speed data concentration measurements. The paper is organized as follows. The parameter of interest of Weibull distribution is introduced, and the details of all proposed methods are described in the section “Materials & Methods”. Numerical results are reported in the next section. In the application section, wind speed data from Khiri Rat Nikhom, Koh Samui, and Kanchanadit in Surat Thani province, Thailand are used to illustrate. Finally, a discussion and conclusions are provided in the last section.

Materials & Methods

A flowchart of the research methodology is shown in Fig. 1.

Figure 1 Flowchart of the research methodology.

Notations and Abbreviations.		
GCI	Generalized confidence interval	
MOVER	Method of variance estimates recovery	
HPD	Highest posterior density	
MLEs	Maximum likelihood estimators	
AIC	Akaike Information Criterion	
MCMC	Markov chain Monte Carlo	
RWM	Random walk Metropolis	
X i	random variables	
c i	scale parameter	
k i	shape parameter	
n i	number of sample size	
p	number of sample case	
μ i	means	
μ ˆi	estimator of means	
μ ˆ	estimator of the common mean	
μ ˆt	Bayesian estimator of the common mean	
var ˆμ ˆi	variance estimate of μ ˆi	
Γ(r)	gamma function; Γ(r) = (r − 1)!	
R k i	Generalized pivotal quantity of ki	
R c i	Generalized pivotal quantity of ci	
R μ	Generalized pivotal quantity of µ	
f(xij; ci, ki)	Probability density function	
F(xij; ci, ki)	Cumulative distribution function	
L(c′, k|x)	Likelihood function	
π(c′, k|x)	Posterior density function of c′ and k	
π(k|c′, x)	Conditional posterior distribution of k	
π(c′|k, x)	Conditional posterior distribution of c′	
π(k)	Prior distribution of k	
π(c′)	Prior distribution of c′	

Suppose that Xi = (Xi1, Xi2, …, Xini) are random variables from Weibull populations with size ni, scale parameters ci, shape parameters ki, and probability density function (pdf) (1) fxij;ci,ki=kicixijciki−1 exp−xijciki,xij>0,

for i = 1, 2, …, p and j = 1, 2, …, ni. The cumulative distribution function is defined by (2) Fxij;ci,ki=1−exp−xijciki.

The parameters ci and ki were estimated based on the maximum likelihood estimation. The maximum likelihood estimators (MLEs) of the two parameters must always be computed numerically. The MLE k ˆi of ki is solution of the following equation (3) 1k ˆi− ∑xijk ˆi lnxij∑xijk ˆi+1ni ∑lnxij=0,

and the MLE c ˆi of ci is given by (4) ci ˆ=∑xijk ˆi/ni1k ˆi.

Consider p independent Weibull populations, the means for which can be derived as (5) μi=ciΓ1+1ki.

Thus, the estimator of μi can be approximated as (6) μ ˆi=c ˆiΓ1+1k ˆi,

where Γ is a gamma function used as an extension of the factorial function for nonintegral numbers. For positive number r, the gamma function can be defined as (7) Γr= ∫0∞xr−1e−xdx=r−1!

An approximation approach can be applied to determine the variance of an estimator. A delta method is a well-known approach for estimating the variance of μ ˆi as follows: (8) var ˆμ ˆi=var ˆk ˆi∂μi∂ki|c ˆi,k ˆi2+2Cov ˆk ˆi,c ˆi∂μi∂ki|c ˆi,k ˆi∂μi∂ci|c ˆi,k ˆi+var ˆc ˆi∂μi∂ci|c ˆi,k ˆi2.

The formulas for the covariance and variance estimates of ci ˆ and ki ˆ are calculated by using the Fisher information matrix (see Cohen (1965) for more detailed information) as follows:

F−1=−∂2μ∂k2−∂2μ∂c∂k−∂2μ∂c∂k−∂2μ∂c2−1=var ˆk ˆCov ˆk ˆ,c ˆCov ˆk ˆ,c ˆvar ˆc ˆ

According to Graybill & Deal (1959), the estimator for common mean µ is derived by using the weighted average of mean μ ˆi based on p individual samples as follows: (9) μ ˆ= ∑i=1pμ ˆivar ˆμi ˆ/∑i=1p1var ˆμi ˆ,

where μ ˆi is defined as in Eq. (6), and var ˆμ ˆi is the variance estimate of μ ˆi, which is defined in Eq. (8).

Generalized Confidence Interval

Weerahandi (1993) introduced the GCI based on the concept of GPQ. Let X = (X1, X2, …, Xn) be a random variable from a distribution with probability density function, which depends on a parameter of interest φ, and a nuisance parameter γ. And let x = (x1, x2, …, xn) be the observed value of random variables X. R(X; x, φ, γ) is called the GPQ if the following two properties hold. These are the distribution of the random quantity R(X; x, φ, γ) is free of unknown parameters, and the observed value r(X; x, φ, γ) do not depend on nuisance parameters. Then, if R(X; x, φ, γ) satisfies the two properties, the quantiles of R form a (1 − α) confidence interval. Now, let Rφ(α) be the α-th quantile of R(X; x, φ, γ). Hence, the 100 α-th two-sided GCIs for the parameter of interest is [Rφ(α/2), Rφ(1 − α/2)].

Let c ˆi0 and k ˆi0 be the observed values of c ˆi and k ˆi based on a sample of size ni from Weibull(ci, ki). Using the results from Thoman, Bain & Antle (1969), the distributions of k ˆiki and k ˆilnc ˆici do not depend on c and k. Consequently, we see that k ˆiki∼k ˆi∗ and k ˆilnc ˆici∼k ˆi∗lnc ˆi∗, where c ˆi∗ and k ˆi∗ are the MLEs based on a sample of size ni from Weibull(1, 1). The GPQs of shape and scale parameters from Weibull distributions were given in Krishnamoorthy, Mukherjee & Guo (2007). (10) Rki=kik ˆik ˆi0=k ˆi0k ˆi∗,i=1,2,…,p,

and (11) Rci=cic ˆik ˆik ˆi0c ˆi0=1c ˆi∗k ˆi∗k ˆi0c ˆi0,i=1,2,…,p.

The GPQ for estimating µ based on the i − th sample is determined as (12) Rμi=RciΓ1+1Rki.

The GPQ for the common mean is a weighted average of the GPQ Rμi based on p individual sample as (13) Rμ= ∑i=1pRμiRvar ˆμi ˆ/∑i=1p1Rvar ˆμi ˆ,

where Rvar ˆμ ˆi is a GPQ of var ˆμ ˆi.

As a result, the 100 (1 − α)% two-sided confidence interval for the common mean using GCI is (14) CIGCI.μ=LGCI.μ,UGCI.μ=Rμα/2,Rμ1−α/2,

where Rμ(α/2) is the 100 α/2-th percentile of Rμ.

The following algorithm is used to construct LGCI.μ and UGCI.μ.

Algorithm 1	
For g = 1 to m, where m is the number of generalized computation	
1. Generate data Xi1∗,Xi2∗,…,Xini∗ from Weibull(1, 1)	
2. Compute c ˆi∗ and k ˆi∗	
3. Compute GPQ of ki, Rki from Eq. (10)	
4. Compute GPQ of ci, Rci from Eq. (11)	
5. Compute GPQ of μi, Rμi from Eq. (12)	
6. Compute GPQ of µ, Rμ from Eq. (13)	
End g loop	
7. Compute Rμ(α/2) and Rμ(1 − α/2)	

Adjusted method of variance estimates recovery

Based on MOVER originally introduced by Donner & Zou (2010), we used it with the large sample method to estimate adjusted MOVER. Again, the pooled estimator of the common mean can be defined as in Eq. (9). μ ˆ= ∑i=1pμ ˆivar ˆμi ˆ/∑i=1p1var ˆμi ˆ.

Consider two parameters of interest μ1 and μ2 with μ ˆ1 and μ ˆ2 as their respective estimators. Assuming that μ ˆ1 and μ ˆ2 are independent, then lower limit L and the upper limit U for μ ˆ1+μ ˆ2 can be defined as (15) L,U=μ ˆ1+μ ˆ2±zα/2Vμ ˆ1+Vμ ˆ2,

where zα/2 is the 100 (α/2) − th percentile of the standard normal distribution.

By using the central limit theorem, the variance estimates for μ ˆi at μi = li, i =1 , 2 are given by (16) V ˆμ ˆl1=μ ˆ1−l12zα/22,

and (17) V ˆμ ˆl2=μ ˆ2−l22zα/22,

where l1 and l2 are the lower limits of μ1 and μ2, respectively.

Furthermore, the variance estimates for μ ˆi at μi = ui, i =1 , 2 are given by (18) V ˆμ ˆu1=u1−μ ˆ12zα/22,

and (19) V ˆμ ˆu2=u2−μ ˆ22zα/22,

where u1 and u2 are the upper limits of μ1 and μ2, respectively.

Based on Eqs. (15)– (19), the 100(1 − α)% confidence limit for μ ˆ1+μ ˆ2 can be written as (20) L=μ ˆ1+μ ˆ2−μ ˆ1−l12+μ ˆ2−l12,

and (21) U=μ ˆ1+μ ˆ2+u1−μ ˆ12+u2−μ ˆ22.

For p independent samples to which adjusted MOVER is applied, lower limit L and upper limit U for the sum of μi can be written as (22) L,U=μ ˆ1+…+μ ˆp±zα/2Vμ ˆ1+…+Vμ ˆp.

The variance estimates of μ ˆi at μi = li and μi = ui, where i = 1, 2, …, p are given by (23) V ˆμ ˆli=μ ˆi−li2zα/22,

and (24) V ˆμ ˆui=ui−μ ˆi2zα/22.

In the present study, the lower and upper limits of μ ˆi are applied based on the Wald confidence interval as follows: (25) li,ui=explnμ ˆi−zα/2var ˆlnμ ˆi,explnμ ˆi+zα/2var ˆlnμ ˆi.

When using the large sample concept to perform interval estimation for µ, the variance estimate of μ ˆi can be defined as (26) var ˆwμ ˆi=12V ˆμ ˆli+V ˆμ ˆui=12μ ˆi−li2zα/22+ui−μ ˆi2zα/22.

Therefore, the 100(1 − α)% two-sided confidence interval for the common mean using the Adjusted MOVER with the Wald confidence interval becomes (27) CIAM.μ=LAM.μ,UAM.μ.

(28) LAM.μ=μ ˆ−zα/21∑i=1p1/V ˆμ ˆli=μ ˆ−1∑i=1p1/μ ˆi−li2,

and (29) UAM.μ=μ ˆ+zα/21∑i=1p1/V ˆμ ˆui=μ ˆ+1∑i=1p1/ui−μ ˆi2,

where μ ˆi is defined as in Eq. (6).

The following algorithm is used to construct LAM.μ and UAM.μ.

Algorithm 2	
1. Compute μ ˆi from Xi	
2. Compute μ ˆ from Eq. (9)	
3. Compute lower and upper limits of μ ˆi from Eq. (25)	
4. Compute LAM.μ from Eq. (28)	
5. Compute UAM.μ from Eq. (29)	

Bayesian Confidence Interval

Bayesian methodology is based on Baye’s theorem for updating the probability based on prior knowledge. The posterior probability is first obtained by using a prior probability distribution and a likelihood function. Here, Bayesian methods for establishing the confidence interval for the common mean of several Weibull distributions are presented. Assume X is a random variable with a Weibull distribution. If c′=1ck, then the pdf can be expressed as (30) fx;c′,k=c′kxk−1 exp−c′xk,x>0.

A Bayesian confidence interval estimate is constructed based on the posterior distribution, a conditional distribution derived from the observed sample data that is used to gain information about the parameter, which is regarded as a random quantity. This is achieved in accordance with the relationship posterior distribution ∝ prior distribution × likelihood function. Hence, we have to provide a suitable prior distribution and likelihood function. In this study, we assumed that the shape and scale parameters follow the gamma prior distribution; i.e., (31) πk∼gammav1,z1,

and (32) πc′∼gammav2,z2,

where v1, z1, v2, z2 are the hyperparameters. As a consequence, the joint posterior density function of c′ as well as k given x can indeed be printed as (33) πc′,k|x∝πc′πk×Lc′,k|x,

and the likelihood function L(c′, k|x) is given by (34) Lc′,k|x= ∏c′kxk−1 exp−c′xk.

For Weibull distribution, π(c′, k|x) cannot be obtained in close form, we used a Gibbs sampling procedure, the Markov chain Monte Carlo (MCMC) method introduced by Geman & Geman (1984), to generate a sample from the posterior density function. The MCMC method is widely used for Bayesian computation in complex statistical models. It generates the samples by rolling a properly constructed Markov chain for an extended period of time. The Gibbs sampler requires samples from fully conditional distributions, which is computationally intensive. The respective conditional posterior distributions of the shape and scale parameters are (35) πk|c′,x∝kn+v1−1 exp−kv1−c′∑xk,

and (36) πc′|k,x∼gamman+v2,z2+ ∑xk.

Although we used Gibbs’ sampling directly for the conditional posterior distribution of the scale parameter, the conditional posterior distribution of the shape parameter does not have a closed form, so Gibbs’ sampling could not be applied in a straightforward manner. Therefore, the Random Walk Metropolis (RWM) algorithm was utilized to generate random samples from an unknown distribution. Similar to acceptance-rejection sampling, the algorithm requires that the applied value has an acceptable probability for each iteration of the algorithm to ensure that the Markov chain converges for the goal density (Saraiva & Suzuki, 2017). To use the RWM algorithm to update the shape parameter, the updated value is approved with probability min(1, Ak), where Ak is defined by (37) Ak=Lk ~,c′|xπk ~Lk,c′|xπk,

where c′(t) and k(t), t =1 , 2, …, T are the Bayesian estimators of c′ and k based on Gibbs’ sampling, respectively. Subsequently, we used the following algorithms to generate the samples and compute the Bayesian estimates.

Algorithm 3 The Gibbs algorithm	
1. Consider the initial state of parameter (c′(0), k(0)).	
For t = 1 to T, where T is the number of iterations using MCMC by Gibbs sampling	
2. Generate c′(t) ∼ gamma(n + v2, z2 + ∑xk(t−1))	
3. Update k(t) using RWM algorithm	
End t loop	
4. Discard the first 1,000 samples	

Algorithm 4 RWM	
1. The initial state of (c′(t), k(t−1))	
2. Generate ɛ from Normal distribution with parameter 0,σk2	
3. Calculate k ~=kt−1+ɛ	
4. Calculate Ak as given in Eq. (37)	
5. Generate u from Uniform distribution with parameter (0, 1)	
6. Set kt=k ~, if u ≤ min(1, Ak), else set k(t) = k(t−1)	

Again, let Xi = (Xi1, Xi2, …, Xini) be a random sample from Weibull distribution with size ni, scale parameter ci and shape parameter ki. The pooled estimator for the common mean based on the Bayesian method is (38) μ ˆt= ∑i=1pμ ˆitvar ˆμ ˆit/∑i=1p1var ˆμ ˆit,i=1,2,…,p,t=1,2,…,T,

where var ˆμ ˆit are the variance estimates of μ ˆit, which is obtained from Eq. (8). After computing the Bayesian estimates by following Algorithms 3 and 4, and Eq. (38), the confidence interval for µ can be constructed.

Therefore, the 100 (1 − α)% two-sided confidence interval for the common mean using the Bayesian method is given by (39) CIB.μ=LB.μ,UB.μ,

where LB.μ and UB.μ are the lower bound and upper bound of the 100 (1 − α)% equitailed confidence interval and the HPD interval of µ, respectively.

The HDInterval package in the R programming suite was used to compute the HPD interval. The assumption for HPD is that all the values inside the HPD interval have a higher probability density than any outside of it, and thus include the most credible value (Kruschke, 2015). In addition, it gives the narrowest length of the interval in the domain of the posterior probability distribution.

The following algorithm is used to construct LB.μ and UB.μ.

Algorithm 5	
1. Compute c′ ˆt and k ˆt from Algorithm 3	
2. Compute μ ˆt from Eq. (38)	
3. Construct the 95% equitailed confidence interval and HPD interval for µ using Eq. (39)	

Results

A simulation study was conducted using the R statistical package. The coverage probabilities and expected lengths of the confidence interval methods were used to evaluate their performance. The data were generated from several independent Weibull distributions denoted as Weibull(ci, ki) where ki = 2 and ci=μ/Γ1+1ki, for i = 1, 2, …, p; common mean μ = 0.5, 1, 5,  or 10; the number of samples p = 2, 4,  or 6; and sample sizes ni for which are provided in Tables 1–3. For each set of parameters, we conducted 5,000 simulation runs, 2,500 pivotal quantities for GCI, and 20,000 MCMC realizations using Gibbs sampling with a burn-in of 1,000 for the Bayesian methods. The method with a coverage probability above the nominal confidence level of 0.95 and the shortest expected length was considered the best-performing one for each scenario. The simulation results for p = 2, 4,  and 6 are reported in Tables 1–3, respectively. Figure 2 shows the algorithm utilized to help estimate the coverage probabilities and expected lengths of the methods.

Table 1 Comparison results of the 95% confidence intervals for the common mean of several Weibull distributions for p = 2.

n	µ	Coverage probability (Expected length)	
		GCI	AM	Equitailed	HPD	
102	0.5	0.9382 (0.2531)	0.8530 (0.1944)	0.9428 (0.2485)	0.9394 (0.2470)	
	1	0.9512 (0.5104)	0.8636 (0.3929)	0.9444 (0.4849)	0.9366 (0.4819)	
	5	0.9458(2.5280)	0.8564 (1.9456)	0.9388 (2.4279)	0.9322 (2.4125)	
	10	0.9440 (5.0889)	0.8586 (3.9137)	0.9402 (4.9150)	0.9316 (4.8824)	
10,20	0.5	0.9495 (0.2033)	0.8850 (0.1663)	0.9552 (0.2009)	0.9495 (0.1996)	
	1	0.9492 (0.4048)	0.8740 (0.3301)	0.9422 (0.3929)	0.9386 (0.3904)	
	5	0.9494 (2.0277)	0.8816 (1.6591)	0.9456 (1.9807)	0.9378 (1.6975)	
	10	0.9475 (4.0627)	0.8800 (3.3107)	0.9412 (3.9779)	0.9335 (3.9512)	
202	0.5	0.9488 (0.1737)	0.9060 (0.1476)	0.9534 (0.1718)	0.9486 (0.1711)	
	1	0.9532 (0.3467)	0.9056 (0.2941)	0.9514 (0.3392)	0.9464 (0.3377)	
	5	0.9498 (1.7309)	0.9030 (1.4764)	0.9452 (1.7021)	0.9396 (1.6946)	
	10	0.9558 (3.4724)	0.9050 (2.9559)	0.9498 (3.4234)	0.9478 (3.4082)	
10,50	0.5	0.9615 (0.1412)	0.9220 (0.1219)	0.9600 (0.1397)	0.9595 (0.1390)	
	1	0.9570 (0.2834)	0.9110 (0.2444)	0.9530 (0.2793)	0.9508 (0.2779)	
	5	0.9476 (1.6690)	0.9092 (1.2023)	0.9489 (1.5734)	0.9481 (1.5689)	
	10	0.9464 (2.8311)	0.9036 (2.4481)	0.9446 (2.0847)	0.9418 (2.7905)	
20,50	0.5	0.9597 (0.1241)	0.9277 (0.1106)	0.9580 (0.1233)	0.9557 (0.1227)	
	1	0.9490 (0.2477)	0.9175 (0.2210)	0.9470 (0.2449)	0.9457 (0.2438)	
	5	0.9512 (1.2406)	0.9257 (1.1089)	0.9485 (1.2291)	0.9462 (1.2234)	
	10	0.9467 (2.4793)	0.9115 (2.2078)	0.9455 (2.4602)	0.9425 (2.4489)	
502	0.5	0.9528 (0.1065)	0.9282 (0.0969)	0.9528 (0.1059)	0.9452 (0.1056)	
	1	0.9524 (0.2121)	0.9250 (0.1928)	0.9520 (0.2103)	0.9478 (0.2097)	
	5	0.9532 (1.0648)	0.9294 (0.9668)	0.9518 (1.0574)	0.9497 (1.0543)	
	10	0.9440 (2.1286)	0.9196 (1.9363)	0.9436 (2.1161)	0.9386 (2.1097)	
50,100	0.5	0.9420 (0.0859)	0.9225 (0.0794)	0.9545 (0.0856)	0.9525 (0.0852)	
	1	0.9480 (0.1716)	0.9255 (0.1587)	0.9465 (0.1705)	0.9470 (0.1698)	
	5	0.9360 (0.8608)	0.9185 (0.7966)	0.9355 (0.8566)	0.9345 (0.8531)	
	10	0.9402 (1.3567)	0.9118 (1.3012)	0.9399 (1.3509)	0.9356 (1.3502)	
1002	0.5	0.9552 (0.0740)	0.9372 (0.0691)	0.9540 (0.0737)	0.9534 (0.0735)	
	1	0.9520 (0.1482)	0.9350 (0.1385)	0.9516 (0.1475)	0.9488 (0.1471)	
	5	0.9524 (0.7390)	0.9320 (0.6915)	0.9506 (0.7367)	0.9492 (0.7348)	
	10	0.9441 (1.4235)	0.9343 (1.3928)	0.9478 (1.4211)	0.9472 (1.4208)	
Notes.

102 stands for (10, 10). Bold values denote the coverage probability higher than the nominal confidence level and the shortest expected length.

Table 2 Comparison results of the 95% confidence intervals for the common mean of several Weibull distributions for p = 4.

n	µ	Coverage probability (Expected length)	
		GCI	AM	Equitailed	HPD	
104	0.5	0.9252 (0.1850)	0.8316 (0.1334)	0.9498 (0.1834)	0.9442 (0.1828)	
	1	0.9308 (0.3690)	0.8386 (0.2663)	0.9316 (0.3540)	0.9266 (0.3528)	
	5	0.9220 (1.8568)	0.8294 (1.3312)	0.9120 (1.7956)	0.9060 (1.7892)	
	10	0.9266 (3.6970)	0.8342 (2.6697)	0.9186 (3.5896)	0.9108 (3.5772)	
102, 202	0.5	0.9320 (0.1490)	0.8545 (0.1146)	0.9485 (0.1472)	0.9430 (0.1466)	
	1	0.9365 (0.2976)	0.8670 (0.2294)	0.9375 (0.2892)	0.9305 (0.2880)	
	5	0.9240 (1.4913)	0.8715 (1.1435)	0.9190 (1.4583)	0.9145 (1.4518)	
	10	0.9365 (3.0107)	0.8610 (2.3123)	0.9310 (2.9476)	0.9250 (2.9346)	
204	0.5	0.9228 (0.1262)	0.8830 (0.1025)	0.9372 (0.1250)	0.9332 (0.1247)	
	1	0.9356 (0.2518)	0.8906 (0.2048)	0.9390 (0.2469)	0.9352 (0.2462)	
	5	0.9264 (1.2643)	0.8838 (1.0276)	0.9210 (1.2438)	0.9172 (1.2404)	
	10	0.9278 (2.5146)	0.8850 (2.0498)	0.9252 (2.4798)	0.9222 (2.4729)	
102, 502	0.5	0.9375 (0.1035)	0.8955 (0.0857)	0.9475 (0.1019)	0.9480 (0.1015)	
	1	0.9490 (0.2053)	0.9060 (0.1705)	0.9490 (0.2023)	0.9470 (0.2015)	
	5	0.9355 (1.0345)	0.8960 (0.8533)	0.9325 (1.0234)	0.9325 (1.0191)	
	10	0.9325 (2.0581)	0.8920 (1.7011)	0.9235 (2.0369)	0.9215 (2.0284)	
202, 502	0.5	0.9396 (0.0892)	0.9223 (0.7760)	0.9476 (0.0886)	0.9505 (0.0882)	
	1	0.9446 (0.1784)	0.9193 (0.1549)	0.9476 (0.1764)	0.9446 (0.1757)	
	5	0.9457 (0.8112)	0.9202 (0.7334)	0.9466 (0.8035)	0.9454 (0.8010)	
	10	0.9476 (1.7894)	0.9260 (1.5547)	0.9450 (1.7751)	0.9426 (1.7682)	
504	0.5	0.9436 (0.0762)	0.9238 (0.0680)	0.9502 (0.0758)	0.9494 (0.0756)	
	1	0.9426 (0.1527)	0.9258 (0.1362)	0.9440 (0.1513)	0.9410 (0.1510)	
	5	0.9482 (0.7629)	0.9276 (0.6798)	0.9468 (0.7579)	0.9438 (0.7560)	
	10	0.9402 (1.5241)	0.9204 (1.3603)	0.9396 (1.5150)	0.9380 (1.5113)	
502, 1002	0.5	0.9540 (0.0614)	0.9360 (0.0560)	0.9550 (0.0611)	0.9550 (0.0609)	
	1	0.9465 (0.1226)	0.9335 (0.1121)	0.9475 (0.1219)	0.9465 (0.1215)	
	5	0.9485 (0.6153)	0.9330 (0.5614)	0.9490 (0.6119)	0.9460 (0.6096)	
	10	0.9412 (1.2574)	0.9336 (1.1556)	0.9482 (1.2544)	0.9479 (1.2539)	
1004	0.5	0.9482 (0.0527)	0.9380 (0.0488)	0.9510 (0.0525)	0.9496 (0.0520)	
	1	0.9452 (0.1054)	0.9328 (0.0976)	0.9462 (0.1049)	0.9444 (0.1047)	
	5	0.9452 (0.5270)	0.9330 (0.4877)	0.9438 (0.5251)	0.9430 (0.5239)	
	10	0.9516 (1.0576)	0.9206 (0.9843)	0.9344 (1.0556)	0.9415 (1.0550)	
Notes.

104 stands for (10, 10, 10, 10). Bold values denote the coverage probability higher than the nominal confidence level and the shortest expected length.

Table 3 Comparison results of the 95% confidence intervals for the common mean of several Weibull distributions for p = 6.

n	µ	Coverage probability (Expected length)	
		GCI	AM	Equitailed	HPD	
106	0.5	0.9028 (0.1565)	0.8186 (0.1074)	0.9424 (0.1549)	0.9398 (0.1544)	
	1	0.8974 (0.3117)	0.8088 (0.2147)	0.9034 (0.2991)	0.8990 (0.2983)	
	5	0.9084 (1.5608)	0.8190 (1.0768)	0.8968 (1.5088)	0.8924 (1.5043)	
	10	0.9042 (3.1252)	0.8236 (2.1468)	0.8904 (3.0312)	0.8842 (3.0222)	
103, 203	0.5	0.9125 (0.1249)	0.8580 (0.0928)	0.9420 (0.1230)	0.9395 (0.1225)	
	1	0.9380 (0.1707)	0.9115 (0.1390)	0.9390 (0.1682)	0.9400 (0.1675)	
	5	0.9015 (1.2454)	0.8510 (0.9273)	0.8955 (1.2152)	0.8955 (1.2102)	
	10	0.914 (3.0203)	0.9002 (2.0118)	0.9045 (2.9023)	0.9056 (2.9011)	
206	0.5	0.9198 (0.1045)	0.8890 (0.0832)	0.9394 (0.1035)	0.9372 (0.1035)	
	1	0.9194 (0.2093)	0.8870 (0.1664)	0.9258 (0.2051)	0.9206 (0.2046)	
	5	0.9146 (1.0454)	0.8816 (0.8329)	0.9108 (1.0284)	0.9094 (1.0259)	
	10	0.9172 (2.0924)	0.8836 (1.6653)	0.9112 (2.0634)	0.9084 (2.0583)	
103, 503	0.5	0.9320 (0.0858)	0.8995 (0.0693)	0.9480 (0.0841)	0.9455 (0.0837)	
	1	0.9225 (0.1709)	0.8925 (0.1388)	0.9290 (0.1683)	0.9270 (0.1676)	
	5	0.9350 (0.8562)	0.8965 (0.6932)	0.9335 (0.8465)	0.9340 (0.8429)	
	10	0.9425 (2.0648)	0.9025 (1.7096)	0.9375 (2.0425)	0.9360 (2.0341)	
203, 503	0.5	0.9355 (0.0734)	0.9235 (0.0632)	0.9490 (0.0729)	0.9490 (0.0726)	
	1	0.9315 (0.1469)	0.9115 (0.1264)	0.9335 (0.1453)	0.9295 (0.1448)	
	5	0.9378 (0.8126)	0.9145 (0.6566)	0.9388 (0.8100)	0.9356 (0.7989)	
	10	0.9412 (1.3555)	0.9243 (1.1923)	0.9456 (1.3510)	0.9434 (1.3502)	
506	0.5	0.9396 (0.0625)	0.9258 (0.0553)	0.9474 (0.0622)	0.9450 (0.0621)	
	1	0.9328 (0.1251)	0.9256 (0.1105)	0.9372 (0.1240)	0.9364 (0.1237)	
	5	0.9338 (0.6258)	0.9232 (0.5534)	0.9330 (0.6218)	0.9308 (0.6203)	
	10	0.9364 (1.2508)	0.9216 (1.1075)	0.9342 (1.2441)	0.9320 (1.2412)	
503, 1003	0.5	0.9455( 0.0503)	0.9380 (0.0457)	0.9555 (0.0502)	0.9540 (0.0500)	
	1	0.9470 (0.1005)	0.9295 (0.0913)	0.9455 (0.0999)	0.9455 (0.0995)	
	5	0.9355 (0.5028)	0.9310 (0.4568)	0.9370 (0.5002)	0.9330 (0.4983)	
	10	0.9403 (0.9148)	0.9345 (0.8292)	0.9389 (0.9045)	0.9388 (0.9028)	
1006	0.5	0.9460 (0.0431)	0.9384 (0.0397)	0.9502 (0.0430)	0.9480 (0.0429)	
	1	0.9430 (0.0862)	0.9354 (0.0795)	0.9496 (0.0859)	0.9474 (0.0857)	
	5	0.9460 (0.4315)	0.9334 (0.3982)	0.9440 (0.4298)	0.9416 (0.4288)	
	10	0.9471 (0.8117)	0.9352 (0.7813)	0.9467 (0.8095)	0.9431 (0.8078)	
Notes.

106 stands for (10, 10, 10, 10, 10, 10). Bold values denote the coverage probability higher than the nominal confidence level and the shortest expected length.

Figure 2 The algorithm for estimating the coverage probability and expected length.

For p = 2, the coverage probabilities calculated using the GCI method were larger than or close to the nominal confidence level for all sample sizes. The Bayesian two-tailed credible interval method was satisfactory in most cases while the Bayesian HPD method only performed well when μ = 0.5 or 1 for large sample sizes. Nevertheless, those using the adjusted MOVER method did not meet the goal in any situation. For p = 4, the coverage probabilities using the GCI method were slightly smaller than 0.95 but performed better with larger sample sizes whereas adjusted MOVER still performed badly. Meanwhile, the Bayesian methods had coverage probabilities higher than 0.95 only when μ = 0.5 for large sample sizes. Moreover, similar results were obtained for p = 6. Finally, the coverage probabilities and expected lengths of the proposed methods for circumstances with varying sample cases, sample sizes, and common mean, are summarized in Figs. 3–5, respectively.

Figure 3 Performance comparison of the methods according to the number of samples ( p) in terms of their (A) coverage probabilities and (B) expected lengths.

Figure 4 Performance comparison of the methods according to the sample size ( n) in terms of their (A) coverage probabilities and (B) expected lengths.

Figure 5 Performance comparison of the methods according to the common mean (µ) in terms of their (A) coverage probabilities and (B) expected lengths.

Application of the Methods to Estimate Wind Speed Data from Various Areas of Surat Thani

Surat Thani is the largest province in southern Thailand and is located on the west coast of the Gulf of Thailand. Ten years of monthly wind speed data were obtained from weather stations in three districts: Khiri Rat Nikhom, Koh Samui, and Kanchanadit (2010-2019) by the Thailand Meteorological Department (Table 4). The data summary statistics are displayed in Table 5. First, we used the Akaike Information Criterion (AIC) to check whether the Weibull distribution was appropriate for these datasets, with the results in Table 6 showing that this was indeed the case with the smallest AIC value. Moreover, Fig. 6 exhibits Q-Q plots of the datasets showing that the Weibull distribution is definitely appropriate, and confirming with P-value of these areas are 0.3708, 0.4826, and 0.5681, respectively. The estimated common mean of the datasets is 0.8869. The 95% interval estimation results for the common mean computed by using all four methods are summarized in Table 7. Furthermore, a trace plot of the generated µ values is shown in Fig. 7.

Table 4 Monthly wind speed data from three areas in Surat Thani.

	Monthly wind speed data (m/s)	
	1.0289	1.0289	1.1832	0.9774	0.9260	0.6688	0.7717	0.7717	0.8231	0.8746	
	1.0803	0.8746	1.0803	0.9260	1.0289	0.8231	0.6688	0.9774	1.2347	0.7717	
Khiri Rat Nikhom	0.9774	0.7202	1.1832	1.0289	0.7717	0.6688	0.9774	0.8746	1.0803	1.0803	
	0.7717	0.8746	0.8746	0.8746	0.6688	0.5659	0.9774	0.6173	0.7202	0.6173	
	0.8231	0.7202	0.5659	0.5144	0.5144	0.6173	0.6688	0.5659	0.6173	0.4116	
	0.9774	0.7717	0.5659	0.5659	1.2347	1.2347	1.3376	0.9260	0.9774	1.0289	
	0.5659	0.7717	0.4630	0.5144	1.3890	1.2861	1.3376	1.0289	1.2347	1.1318	
Koh Samui	0.6173	0.9774	0.9260	0.8746	1.4919	1.2347	1.6462	1.1832	1.2347	1.1318	
	0.2572	0.7202	0.8231	0.9774	1.2347	1.3890	1.7491	1.2347	1.2347	1.1832	
	0.6688	1.2347	0.8746	1.0803	1.2861	1.1318	1.8520	1.0803	1.2347	1.3376	
	0.2572	0.3601	0.6173	1.0289	0.9260	0.4116	0.9774	0.8746	0.9260	0.8746	
	0.2572	0.3601	0.5659	0.8231	1.1318	0.7202	0.8231	1.1318	1.3890	0.8231	
Kanchanadit	0.2572	0.3601	0.7202	0.8231	0.7202	0.5659	1.1832	1.0289	1.0289	1.2861	
	0.1029	0.4630	0.7202	0.6173	0.6173	0.6173	1.2347	0.8231	0.4630	0.5659	
	0.1543	0.3087	0.3087	0.3601	0.3087	0.4630	0.8746	0.6173	0.3087	0.2572	

Table 5 Summary statistics for the wind speed data from three areas in Surat Thani.

Areas	n i	c ˆi	k ˆi	μ ˆi	
Khiri Rat Nikhom	50	0.9064	4.7668	0.8299	
Koh Samui	50	1.1804	3.6349	1.0642	
Kanchanadit	50	0.7564	2.2178	0.6699	

Table 6 AIC values of the wind speed datasets from three areas in Surat Thani.

	Methods	
	Weibull	Gamma	Log-normal	Normal	Exponential	Cauchy	
Khiri Rat Nikhom	−17.4248	−16.8662	−15.4456	−17.3832	83.2807	8.1797	
Koh Samui	34.2095	40.3958	46.7676	34.4694	108.2886	49.2098	
Kanchanadit	28.7958	31.0278	36.1130	32.6032	61.7694	58.5561	
Notes.

Bold values denote the smallest AIC value.

Figure 6 Weibull Q-Q plots of the wind speed data from three areas in Surat Thani: (A) Khiri Rat Nikhom, (B) Koh Samui, and (C) Kanchanadit.

Table 7 The 95% interval estimation for the common mean of the wind speed data from three areas in Surat Thani.

Method	Confidence intervals for µ	
	Lower	Upper	Length	
GCI	0.7812	0.9044	0.1232	
AM	0.8502	0.9266	0.0764	
Equitailed	0.7850	0.9040	0.1190	
HPD	0.7820	0.9007	0.1186	

Figure 7 Common mean (µ) vs. the number of iterations using the MCMC algorithm.

It can be seen that these findings confirm the simulation results for a large sample size. Adjusted MOVER provided the shortest expected length, while that of the Bayesian HPD interval was smaller than those of GCI and the Bayesian equitailed confidence interval. However, once again adjusted MOVER yielded a coverage probability that was lower than the other methods and did not reach the target. Meanwhile, both Bayesian methods yielded coverage probabilities higher than the target and the expected length of the Bayesian HPD interval was slightly narrower than that of the Bayesian equitailed confidence interval. Therefore, the Bayesian HPD interval is the most suitable for estimating the confidence interval for the common mean of several Weibull distributions for large sample sizes.

Discussion

La-ongkaew, Niwitpong & Niwitpong (2021) proposed Bayesian methods using the gamma prior for estimating the difference between the parameter values of several Weibull distributions and applied them to wind speed data measured at wind energy stations in Thailand. We extended this idea to construct estimates for the confidence interval for the common mean of several Weibull distributions using GCI, adjusted MOVER, and the Bayesian equitailed confidence interval and HPD interval both based on the gamma prior. The findings demonstrate that the Bayesian HPD interval achieved the best performance when µ was small with large sample sizes when its coverage probability and expected length were both taken into consideration. Furthermore, GCI managed to perform well together with small sample sizes whereas adjusted MOVER did not perform well in any of the scenarios tested.

Increasing the number of sample cases (p) caused the coverage probabilities of the methods to be less than 0.95 and the expected length to decrease. Although the sample sizes increased, the coverage probabilities of all of the proposed methods improved (closer to 0.95), and the expected lengths became narrower. Moreover, for unequal sample sizes, the Bayesian HPD interval yielded a coverage probability greater than or close to 0.95 with the shortest expected length.

We applied the confidence interval estimates for the common mean of three wind speed datasets from Surat Thani, Thailand, for which the Bayesian HPD interval performed the best. Knowledge of predicting the common mean of the wind speed across this large province could help the provincial authorities to prepare for adverse weather events. Since the Bayesian HPD interval performed the best in this scenario, it has the ability to estimate the confidence interval for the common mean of wind speed datasets from many areas provided that they follow Weibull distributions. As mentioned above, the wind speed distribution may differ for a specific site during different months, seasons, and years. The data should be tested for any kind of distribution using any of the criteria presented either AIC, p-value or qq-plot.

Conclusions

We proposed the Bayesian equitailed confidence interval and the HPD interval using the gamma prior for estimating the confidence interval for the common mean of several Weibull distributions and compared their performances with GCI and adjusted MOVER. From the simulation results, both Bayesian methods yielded coverage probabilities greater than or close to the target with shorter expected lengths than the other methods in most cases for p = 2. They performed well in many cases for large sample sizes for p = 4 and 6. Our findings indicate that GCI generally performed well in terms of coverage probability whereas the Bayesian methods performed better than the others when the value of µ was small. Moreover, adjusted MOVER performed poorly in all cases.

We used wind speed data from Surat Thani province, Thailand, to measure the efficiency of the proposed methods. In this case, the Bayesian HPD interval performed the best and can be used to estimate the confidence interval for the common mean of several Weibull distributions for this particular scenario. In future work, we will expand our research to establish simultaneous confidence intervals for the difference between the means of more than two populations.

Supplemental Information

Supplemental Information 1 R code for common mean of Weibul distribution

This main program produced all results in Tables

Click here for additional data file.

Supplemental Information 2 The wind speeds measured at 90-meter wind energy potential stations from four regions in Thailand

Central, Northern, Western and Southern

Click here for additional data file.

Additional Information and Declarations

Competing Interests

Author Contributions

Data Availability

The authors declare there are no competing interests.

Manussaya La-ongkaew conceived and designed the experiments, performed the experiments, analyzed the data, prepared figures and/or tables, authored or reviewed drafts of the article, and approved the final draft.

Sa-Aat Niwitpong conceived and designed the experiments, analyzed the data, authored or reviewed drafts of the article, and approved the final draft.

Suparat Niwitpong performed the experiments, analyzed the data, prepared figures and/or tables, authored or reviewed drafts of the article, and approved the final draft.

The following information was supplied regarding data availability:

The data and code are available in the Supplemental Files.

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
