# Peer review of "Estimating average wind speed in Thailand using confidence intervals for common mean of several Weibull distributions"

_PeerJ, doi:10.7717/peerj.15513_

## Round 0.1 · original submission · Major Revisions

The work is interesting, has a clear degree of originality, and is appropriate for publication in the journal after performing a major and very careful revision. Nevertheless, it needs some further improvements. In general, there are still some occasional grammar errors throughout the manuscript, especially the article "the," "a," and "an" are missing in many places; please make spellchecking in addition to these minor issues. The reviewer has listed some specific comments that might help the authors further enhance the manuscript's quality.

1. Specific Comments

• Overall, the Abstract section is not giving any information about methodology, results, conclusion, and recommendations as it should be. I suggest the authors remove generic lines and present the strong statements and novelty of the article. The abstract is written by qualitative sentences. It needs to be modified and rewritten based on the most important quantity results from this research. The abstract should be redesigned. You should avoid using acronyms in the abstract and insert the work's main conclusion.

• You have used many abbreviations in the text. From this perspective, an Index of Notations and Abbreviations would be beneficial for a better understanding of the proposed work. Furthermore, please check carefully if all the abbreviations and notations considered in the work are explained for the first time when they are used, even if these are considered trivial by the authors. The paper should be accessible to a wide audience. Furthermore, it will make sense to include also the notations in this index.

• The objectives should be more explicitly stated.

• The Introduction section must be written more clearly. The research gap should be delivered in a more clear way with directed necessity for the conducted research work.

• What is the novelty of this work?

• It is better to improve your contributions which are not so clear to show the advantage of your work.

• The novelty of this work must be clearly addressed and discussed in the Introduction section.

• The methodology limitation should be mentioned.
Many equations are presented in the paper, and most look OK. However, please check carefully whether all equations are necessary and whether the quantities involved are properly explained. Also, some equations need references.

• Results
• This section is well written.

• Discussion
• Overall, the discussion part is weak. The Discussion should summarize the manuscript's main finding(s) in the context of the broader scientific literature and address any study limitations or results that conflict with other published work.

• Conclusion
• Some future works should be added to your conclusion.

·

Basic reporting

In Abstract section, the line 18-19 the idea is not very clear: The results from a simulation study indicate that Bayesian equitailed confidence interval is appropriate when the value of the common mean is small for a large sample size…………..Q1: Can you please give some specification/indicators or you can write it in more concise way? (give for your case study and then global interest).
I suggest reorganizing the abstract section specifically line 12 up to 19.
Introduction section is constructed in a very interesting way, and authors have used good references (including the newest one, and different case studies from the region). Moreover, the Bayesian method for constructing confidence intervals for the mean of a normal distribution with an unknown coefficient of variation is also used as a reference. As a conclusion this section is well arranged.
Q2: How many years do you have considered performing such measurements?
An accurate estimate of the long-term wind speed statistical characteristics at a potential wind farm site is an essential input to the financial assessment of a project, since the energy production of a wind farm is very sensitive to wind speed. Ideally wind data measurements over a 10-year period are required.
As one of the paper’s task is to estimate the confidence interval of μ. Referring to a “characteristic” data set you assume that the latter is dense-enough and long-enough not to exclude systematic variations of diurnal or seasonal type.
Monte Carlo simulation methods and, in particular, Markov chain Monte Carlo methods, play a large and recognized role in the practice of Bayesian statistics, where these methods are used to summarize the posterior distributions that arise in the context of the Bayesian prior–posterior analysis.
Monte Carlo methods are used in practically all aspects of Bayesian inference, for example, in the context of prediction problems and in the computation of quantities, such as the marginal likelihood, that are used for comparing competing Bayesian models.

Experimental design

Method section is well organized but possibly in the equation 1 I suggest changing the Weibull scale parameter from ai to ci as it is more understandable, and many authors have used this symbol, so far.
Q3: Secondly, kindly include the formula and clarify “The cumulative distribution function” and gamma function (in equation 4). What is gamma function in equation 4… please clarify….. and differences from gamma function that relates the two Weibull parameters and the average wind speed……

Validity of the findings

Data's validity and reliability are key aspects of all research on the wind power technology. Hence, Authors have shown meticulous attention to these two aspects.

Authors have proposed Bayesian methods using a gamma prior distribution for constructing confidence intervals for the common mean of several Weibull distributions and compared their performances with GCI and adjusted MOVER.

Additional comments

The figures should be formatted in a more clearer way ……..and I suggest working with the OriginPRO software or other related software…..

Reviewer 2 ·

Basic reporting

The manuscript deals with the application of the Weibull distribution for wind energy assessment in Thailand. Although - according to the title - the main focus in the manuscript is on wind energy, it starts with a long statistical part, followed by a very meager chapter where the wind energy aspects are presented. There should be a better balance between the two parts.

Experimental design

The manuscript lacks a description of the wind observations, the location of the masts and type of instrumentation. It lacks a description of the individual wind distributions from the masts.

Validity of the findings

It has to beclarified in the manuscript. that the Weibull distribution provides an empirical fit to averaged (typical 10 min) wind speed data and has been shown in many cases to be an appropriate fit but not in all cases.
As I understand the manuscript, all wind data from a region have been lumped together and used to derive the Weibull parameters (A and k) for each of the region. I assume that ni in table 4 represents the number of individual meteorological stations in the region. The wind energy potential depends very much on the local conditions (hills, roughness) and I wonder why a regional averaged wind speed can be of use for wind energy assessment. The authors need to discuss the aspect of local to regional wind energy.
In table 5 the AIC values for the four regions are shown but it is not discussed if one distribution is significantly different from another, it seems that the distribution with the larger number is simply chosen. This raises the question if it is permissible to approximate the observations from region Eastern by a Weibull distribution although the largest number is for the log-normal distribution. Please discuss why you in objective terms have removed this region from the analysis. In table 4 a q-q plot are shown for 3 regions, please add q-q plot for the Eastern region. I note that the q-q plot for region C is poor – please discuss.

Additional comments

Add some recommendations for applied use – when is it permissible to apply an Weibull distribution and when should it not be used – some objective criteria.
I note that parameters throughout the manuscript lack units – I presume that the wind speed is in m/s and not normalized.

·

Basic reporting

1.A flow chart can be designed in the article to help readers better understand the research method, and this flow chart describes the steps the author has taken in the research process. This algorithm has been applied in this field, the most important thing is to list the trend of early development and final goal of the research method in the introduction. Adding a flowchart will help readers understand the approach proposed in this article.

Experimental design

1.The article should supplement the data charts obtained under GCI, MOVER and Bayesian methods for comparison of the conclusions under the three methods, briefly summarize them respectively, add text content, and comment on the results generated under different methods.
2.Gibbs algorithm and Random walk Metropolis algorithm (RWM) are mentioned in Bayesian confidence interval method, which can supplement the relevant contents of Gibbs sampling and RWM algorithm in detail.

Validity of the findings

1.This article estimates the average wind speed in Thailand using the confidence intervals for the common mean of several Weibull distributions. However, when relevant studies show that the location parameter µ value is used to estimate the confidence interval of the mean value of the Weibull distribution, it has a high accuracy in estimating the mean value of the wind speed. Whether to consider the confidence intervals estimated by other parameters in the Weibull distribution, and whether to describe the fitting estimation of other shape parameter k values and scale parameter c values in detail, text content and charts should be added in the article to explain.

Additional comments

1.For the application of this method in estimating wind speed data in various regions of Thailand, each chart should be explained in more detail. At the same time, whether this method is also applicable to other countries and regions should be summarized at the end.
2.The conclusion part can not only summarize the main work of this paper, but also add some supplements or prospects for future work.

---

## Round 0.2 · Major Revisions

Please check the comments of Reviewer 2 carefully.

Reviewer 2 ·

Basic reporting

In the original manuscript, the application of the study was on wind energy. This aspect has been removed in the revised manuscript (I wonder why) and now it is stated in the introduction that the goal of the study is to “predict the occurrence of severe wind speed events” (lines 100 – 101) and in line 14 “leading to catastrophic events”. It is as if the authors have not clarified for themselves the practical and application aspects of their study and only concentrated on the statistical developments. As such the manuscript really belongs to a journal that is purely devoted to statistics.
Furthermore, the study in the revised manuscript is based on a different dataset. These major changes to the manuscript should have been explained in the note to the editor and reviewers when submitting the revised manuscript.

In my review comments it is stated that 4 regions are dealt with in the original manuscript and one of them (Eastern region) is better fitted with a log-normal distribution. I asked the authors to comment on this interesting result but I do not see any reply on this in the answer to my review. It seems, that the authors simply removed such data that did not fit their Weibull distribution assumption – from the analysis. – instead of using the opportunity to discuss the finding. This is not in agreement with good scientific behavior. Actually, I found out that this revised version is based on completely new data-series? This should have been made clear in the reply to the editor and reviewers and it should be made clear why the old data series are not used any more but replaced with other data.

Line 33. Greenhouse gases does not absorb heat, greenhouse gases absorb radiation in certain wavelengths. Furthermore, greenhouse gas does not destroy the ozone layer.

Table 4. Give wind speed in units of m/s.

Figure 6b. The constant values of “sample” Q around 2.5 in Figure 6b suggests that there is something wrong with the observations in the Weibull Q-Q plot of Kob-Samui. Please discuss and explain what sort of quality control of the observations was performed by the authors.

In your answer to my comments. Please make sure to comment on all my questions in a detailed and complete way.

Experimental design

no comments

Validity of the findings

The goal of the study is now “predict the occurrence of severe wind speed events... I would consider severe wind speed events a something that occur a few times each year, and therefore likely cannot be predicted by confidence intervals of a Weibull distribution. I suggest to investigate, how well the highest (say 10 highest) measured wind speeds actually are predicted by the Weibull distributions and report in the manuscript.

Additional comments

no comments

---

## Round 0.3 · accepted · Accept

I congratulate the authors for the effort put into this paper! The manuscript is significantly improved; therefore, I recommend accepting it in its current form!

Reviewer 2 ·

Basic reporting

The revised manuscript basically deals with my comments and can in my opinion be acceptanced.

Experimental design

na

Validity of the findings

na

Additional comments

na